# Twenty Years of Collaboration to Sort out Phage Mu Replication and Its Dependence on the Mu Central Gyrase Binding Site

**DOI:** 10.3390/v15030637

**Published:** 2023-02-27

**Authors:** Ariane Toussaint, N. Patrick Higgins

**Affiliations:** 1Cellular and Molecular Microbiology, Université Libre de Bruxelles (ULB), Rue Adrienne Bolland 8 B, 6041 Gosselies, Belgium; 2Biochemistry and Molecular Genetics, University of Alabama at Birmingham, Birmingham, AL 35294, USA

## Abstract

For 20 years, the intricacies in bacteriophage Mu replication and its regulation were elucidated in collaboration between Ariane Toussaint and her co-workers in the Laboratory of Genetics at the Université Libre de Bruxelles, and the groups of Martin Pato and N. Patrick Higgins in the US. Here, to honor Martin Pato’s scientific passion and rigor, we tell the history of this long-term sharing of results, ideas and experiments between the three groups, and Martin’s final discovery of a very unexpected step in the initiation of Mu replication, the joining of Mu DNA ends separated by 38 kB with the assistance of the host DNA gyrase.

Acting as an Editor for this Special Issue dedicated to phage research in Belgium did not immediately sound like a cup of tea. Despite being officially retired for too long, we could not resist writing this contribution to honor the memory and scientific achievements of a very special friend and collaborator, Martin Pato, who passed away on 7 May 2022. He regularly visited Ariane’s lab at ULB, between the middle 1970s and 1990s, a time when phages were top research models for both genetics and biochemistry in Belgium and all over the world. The publication of the first complete genomic sequence of a virus, phage MS2, by W. Fiers’s group [1] from Ghent in 1976, remains a landmark achievement of that period.

In the early 1970s, several research groups around the world started working on a “weird” mutator phage (Mu), discovered by Larry Taylor [2]. The long-standing collaboration with Martin is an example of how the combination of three scientific manias, phage Mu, DNA replication and DNA supercoiling, led to the uncovering of a novel mechanism of synapsis of the right and left Mu viral ends that are separated by 38 kb of DNA. Ends-pairing initiates the stepwise assembly of the transpososome, a large complex in which the Mu DNA ends are synapsed and nicked by a hexamer of the phage-encoded pA transposase. This complex can launch a concerted nucleophilic attack on the target site for transpositional insertion, followed by disassembly of the transposome and assembly of a complex that replicates Mu DNA, generating two copies of integrated Mu. This process is repeated, amplifying the phage genome, while it remains associated with host DNA (see [3] for a review), until packaging of these integrated copies generates infectious particles.

Ariane and Martin first met in 1974 at an EMBO workshop on restriction enzymes and DNA sequencing. Walter Fiers, Marc van Montagu and Jeff Schell organized this event in a middle-age cloister in Drongen, near Ghent, that was transformed into a Meeting Center. After completing a PhD thesis in Don Glazer’s lab at Berkeley in 1968, Martin worked in the University Institute of Microbiology, in Copenhagen, with K. von Meyenburg and J. Zeuthen. All three were invited to the Drongen meeting. 

During his PhD, Martin identified the origin of *E. coli* chromosomal DNA synthesis and followed the direction of fork movement toward termination [4,5]. He devised clever techniques to study the chemical synthesis and degradation of DNA, RNA and proteins inside living bacterial cells. His interest in the patterns of DNA synthesis expanded to other cellular components as well, leading to an Annual Review of a Microbiology article he wrote in 1972 [6]. In Copenhagen, with von Meyenburg and Zeuthen, Martin analyzed the replication of the F’*lac* sex factor [7] and colicin plasmids [8]. Curious about the in vivo rates of all metabolic processes in bacteria growing under steady-state conditions, they also measured precursor pools and showed that cultures of *E. coli* precisely match the rate of transcription with the rate of protein synthesis [9,10]. This work was years ahead of the critical scientific theory. Later studies proved these observations to be correct. It took years to understand the mechanisms and reasons why transcription/translation rates were so intimately connected. Today, it is understood that these two rates coordinate with protein folding, protein degradation, DNA supercoiling and DNA repair.

Martin returned to the US, accepting a position at the National Jewish Hospital (NJH) in Denver, Colorado, near Norman Pace’s group. At the time, Mu biology and biochemistry were shrouded in mystery. Across the street from NJH, at the University Hospital, Barbara Waggoner was finishing her thesis with Larry Taylor. They characterized, via electron microscopy (EM), unexpectedly large covalently closed circular DNA molecules containing a Mu copy bound to *E.coli* host DNA. These structures accumulated during Mu development [11]. Ariane visited Larry’s lab to learn and transfer knowledge about EM analysis of heteroduplex DNA, because Annette Résibois had joined the Mu group at ULB with her EM expertise, and she also wished to look at images of different Mu DNAs after induction. The big “Mu” question then was what happened to the prophage at the onset of replication. Was it excised as other integrated prophages such as ʎ or did it remain integrated, replicating ‘in situ” Martin and Barbara provided evidence that replicated Mu DNA remained attached to host DNA and that viral replication was confined within viral sequences and did not extend past viral left or right ends [12]. In 1976, Martin, Larry, Ariane and most of the world’s “Muologists” attended a meeting in Cold Spring Harbor, which led to the publication of a book, “DNA Insertion Elements, Plasmids, and Episomes” [13]. The introduction summarized the state of knowledge of “mobile” elements at the time. DNA sequences of defined length and nucleotide sequence, called ISs (Insertion Sequences), were found in *E. coli lac*, *gal* and *mal* operons and in phage ʎ polar mutants. The same IS sequence could appear at different sites in a chromosome. In the F plasmid, ISs were shown to mark sites where the plasmid integrates into the bacterial chromosome. In R plasmids, ISs marked the limit between the plasmid moiety and a ‘resistance module’. Some ISs contained a promoter that could alter expression of genes adjacent to the IS site of insertion. Transposable antibiotic resistance genes had been shown to move from one replicon to another both in Gram^+^ and Gram^−^ bacteria and were found on various plasmids. Some of these properties had also been demonstrated for Mu insertions [14]. In particular, ‘cointegrates’ had been discovered after Mu-promoted insertion in the host chromosome of either F’*lac* or a ʎ derivative deficient for integration and replication [15,16]. These revelations demonstrated new pathways for the evolution of prokaryotic genomes.

One article in the book is of particular interest for this story [17]. By a combination of DNA–DNA hybridization of ^3^H-labeled DNA extracted at various times after the induction of a Mu prophage, DNA isolation on a CsCl gradient and EM analysis, Barbara, Martin and Larry established that (1) newly replicated Mu-specific DNA was present 9 min after induction, and two prophage copies were present at 12 min and six by 15 min; (2) covalently closed circular DNA of heterogenous length (Hc-DNA), not correlated with Mu length, appeared 14 min after induction and diminished in length with time until shortly before lysis. Hc-DNA, thus, appeared after at least one round of prophage replication; (3) formation of Hc-DNA required that the prophage was induced and could replicate and, hence, was correlated with Mu replication.

In the same book, with A.I. Bukhari, the ULB group published a model based on the analysis of Mu-mediated chromosomal rearrangements of host DNA, proposing that Mu replication always proceeds from integrated Mu copies and is systematically accompanied by a chromosomal rearrangement (deletion, inversion, duplication, transposition), a process that generates Hc-DNA [18]. This marked the beginning of the long-term collaboration between Martin and the ULB group, combining expertise in microbial genetics, biochemistry and EM.

Mu ends and pA and pB, the transposase and transposition activator, respectively, are essential for phage DNA replication. To identify other genes or sites essential for replication, the ULB group isolated internal Mu deletions of various lengths, generating so- called mini-Mus. This also facilitated EM analysis of the Mu moiety in Hc-DNA. Mini-Mu replication was tested in Denver by labeling and DNA–DNA hybridization, while EM analysis of DNA accumulating during replication was visualized using EM at ULB. Mini-Mus, deleted of most of the internal region of the phage DNA, but still containing at least the repressor and replication *A* and *B* genes and both termini, replicated upon induction. Deletion of part of gene *B* impaired replication, which was only partially restored when protein B was provided in trans*,* whereas supplying gene *B* and additional functional DNA between genes *B* and *C* allowed for extensive replication of the mini-Mu DNA. Functions, thus, existed between genes *B* and *C*, which amplify the replication of Mu DNA [19], later shown to be the “*arm*” gene product [20].

In 1979, Jim Shapiro published a model for the mechanism of replicative transposition [21]. The model accounted for all Mu-induced rearrangements described by the ULB group. It made specific predictions on the DNA forms that should be observed during Mu or mini-Mu replication, including Hc-DNA, and on the confinement of replication to the transposing Mu DNA. This oriented our EM analysis of the replicating mini-Mu DNA to confirm the existence of the structures predicted by the model. To preserve these structures, which would have been destroyed by heteroduplex mapping, the mini-Mu was localized by its partial denaturation map. This was made possible by the computer skills of Marc Colet, who had recently joined the ULB group [22]. We analyzed the correlation between a standard denaturation map of the mini-Mu and the denaturation profiles of the various DNA structures observed after mini-Mu induction. All the complex structures predicted by Shapiro’s model were observed. In addition, structures were seen, which contained two forks separated by a double-stranded DNA segment shorter than the length of the mini- Mu, hence, possibly a partially replicated mini-Mu, consistent with (1) these DNA structures being mini-Mu replication intermediates and (2) with (mini-)Mu replicating without excision from either or both ends [23].

In 1983, Pat Higgins, then at the University of Wyoming in Laramie, joined the collaboration. He was developing an in vitro replication system for Mu, a good addition to the genetic and biochemical approaches [24]. From then on and until 1993, with travel grants from NATO and NSF in the US and from FNRS in Belgium, we managed to spend one month a year at least in each other’s labs, later joined by PhD students and post-docs.

While we progressed in understanding Mu replication, research on the mechanisms of transposition of IS and Tn elements was going on. In complex transposons containing two IS sequences flanking the antibiotic resistance module (e.g., Tn*9* with IS*1* [25] and Tn*10* with IS*10* [26]), the frequency of transposition decreased when the transposon size increased. In Mu, Martha Howe isolated Tn5 insertions, a 5.8 Kb-long insertion. One of them, Mu*Kn7701*, had the Tn located at a position that should not have blocked replication. Nonetheless, that phage grew very poorly with a long delay in the onset of replication after induction. To further check on this length dependence, Martin suggested isolating “Maxi-Mu’s”. This was readily carried out by Michel Faelen at ULB [27]. The result was a collection of MudI(Ap,lac)-derived prophages, 39.8, 59.0, 85.6 and 88.2 kb-long, respectively. The comparison of these maxi-Mu’s with the 37.2 kb-long parental MudI(Ap,lac) indicated that the transposition frequency decreased as the length of the prophage increased. No replication of the two longest maxi-Mu’s could be detected. The 59 and the 39.8 kb-long chimeric genomes were noted to replicate at approximately 1 to 2 and 30%, respectively, of the rate found with the MudI(Ap,lac) prophage. The length dependance of the transposition and replication could be explained by an impairment in an early step of the transposition/replication mechanism, perhaps the pairing of the two Mu ends.

In 1983, Martin and his technician, Claudia Reich, discovered that Mu transposase was unstable [28] and demonstrated the stoichiometric use of the transposase in rounds following the initial transposition reaction [29]. From 1984 to 1994, the collaboration between Martin, Pat and the ULB lab continued on a range of topics, including transposase instability, further characterization of Mu replication dependence on host proteins and of the Mu repressor Repc. 

In 1985, the Mizuuchi lab published a seminal paper describing a soluble system that would carry out efficient Mu transposition reactions in vitro [30]. They identified essential proteins and enzymes required for both transposition and replication. One requirement was a supercoiled DNA substrate that could be provided with purified DNA gyrase.

At a 1989 Gordon conference, Pat learned an interesting fact about a paper published 7 years earlier in *Virology* by Martha Howe’s lab [31]. Martha had two students that each found one independent rare mutant of Mu they called *nuA* and *nuB*. Both *nu* mutants could form plaques on lawns of mutant *E. coli* strains that wild-type (WT) Mu would not grow on. The manuscript stated that the *E. coli* mutation was in *himA* or *himB* genes, which encode the IHF protein that was essential for Mu induction. The rare *nu* mutations appeared at a frequency estimated between 10^−6^ and 10^−9^ per lysogen and both mutations mapped near the Mu genes *G* and *I,* which are near the center of the viral DNA. At the conference, Jeff Miller told Pat that the *E. coli* mutation was actually in *gyrB* rather than in IHF, and it had a temperature-sensitive phenotype at 42 °C. Pat called Martin to discuss one simple hypothesis: the Mu virus could compensate for a weak gyrase by strengthening a gyrase binding site at its center. From Pat’s background of work on gyrase in the Cozzarelli lab [32,33], he and Martin were poised and ready to roll. Martin requested both *nuA* and *nuB* strains from Martha Howe and she sent both mutants to both labs immediately. Pat purified DNAs from Mu and both *nu* mutants and carried out quinolone-induced gyrase cleavage reactions. A single strong cleavage site appeared on one restriction fragment for all three DNAs. Dramatically, the cleavage efficiency was *nuB* > *nuA* > WT. After seeing the results on a poor fax, Martin flew to Pat’s lab in Birmingham to plan the immediate work, obtain stocks of purified gyrase subunits and special reagents for supercoiling DNA cleavage analyses. The name SGS for "strong gyrase site" was chosen for the site. Returning to Denver with ample socks of critical reagents and gyrase, Martin cloned all three SGS sites into M13 vectors for sequencing and into pBR332 for supercoiling analysis, quinolone cleavage assays and for other purposes yet to be determined. The three sites were named SGS WT, SGSnuB1 and SGSnuB103. Sequencing of cleaved DNA was carried out with the chain termination method and results showed a band in all four sequencing lanes that marked gyrase cleavage sites. Chain-terminating polymerases sometimes add an untemplated nucleotide at the sequence end, providing slightly ambiguous data. With his usual tenacity, Martin wanted to recheck, and he ligated linkers directly onto both ends of the cleaved DNAs and cloned them for sequencing to prove the expected 4 bp-break overhang in each site. Finally, he constructed a Mu prophage variant with a deletion of the SGS. Replication of the deletion mutant was delayed by about 100 min relative to Mu WT. The manuscript was published in PNAS [34]. The gyrase supercoil hypothesis had multiple predictions to test, with a major question being whether the central location between Mu ends was critical. 

In 2004, Martin published a study that proved the center of Mu DNA to be the optimal position for the SGS [35]. As usual, he did this in more than one way. First, the Mu phage with an SGS deletion was genetically modified by integrating a WT SGS at many different locations, from the near left end to near right end. The middle position clearly proved best for Mu replication. Then, he made phage deletions that offset the center between right and left ends. Deletions left and right of the SGS also caused delays in phage development. 

In 1995, a final collaborative paper on the Mu gyrase SGS mechanism was published, reporting on the analysis of the biochemical reaction steps that progressed or were delayed after induction of Mu lysogens that lack the SGS. Transcription and protein production occurred normally. Strikingly, the step of strand cleavage at Mu ends and strand transfer to a new location was delayed by 40 min without the SGS. The critical step promoted by gyrase at SGS was, thus, the assembly of a functional transpososome [36]. This observation, made in vivo on a full-length Mu DNA, was extending to "real life" the in vitro experiments on very short mini-Mus, consisting of just the two ends of the genome, showing that these need to be properly aligned and synapsed in a defined topological way before DNA replication [37,38]. When discussing the requirement for a central location of the SGS in Mu, Martin used to draw the sketch shown in Figure 1. It suggests that the extrusion of a Mu-containing domain within the context of the bacterial nucleoid helps to bring Mu ends together prior to transpososome assembly. 

Martin remained curious about whether any ”strong” gyrase cleavage sites that had been studied by many prior labs could function at the center of Mu. He made multiple attempts to find alternatives from 1999 to 2003 [35,39,40,41]. None of the sites tested promoted Mu replication, with one exception, which was an SGS from the closely related Mu-like prophage in the *E. coli* O157:H7 Sakai genome. Martin’s last paper was published in the *Journal of Bacteriology* in 2006 [42]. He dissected the SGS DNA using deletions coupled with DNase-I foot-printing. Deletions into the left and right arms of the WT SGS showed dramatic differences. The SGS left arm could be deleted or changed with little or no consequence on gyrase cleavage. The right arm, however, was essential, showing strong DNase-I footprint patterns over extended distances. Sequence analysis of the right arm exhibited repeat sequence motifs that induce long-range DNA bending, which promotes strong DNA interaction with one of the GyrA subunit “pinwheel” elements. The consequence is that one DNA strand is passed processively through the gyrase gate for fast and efficient supercoiling [43].

In retrospect, it is curious that all the labs studying single-molecule DNA–Gyrase complexes chose Martin’s pBR322-derived plasmid with the strongest Mu SGSnuB103 sequence to bind gyrase for analysis of discrete steps in a gyrase reaction mechanism. This was used to measure catalytic rates in vitro [44,45]. It may have been a mistake because supercoiling steps at much more abundant sites in the chromosome could follow different and slower paths than supercoiling at the SGS dedicated to mobilizing host gyrase to efficiently replicate Mu DNA, a first step in the production of infectious viral particles.

The work from Martin’s lab provided a significant clue to supercoil research in bacteria. Most well-studied bacteria have negatively supercoiled chromosomes and contain easily recognized genes coding for GyrA and GyrB. Because supercoiling can be influenced both by mutations in gyrase subunits and mutations in high-affinity gyrase binding sites, bacterial species have the potential to optimize the supercoil levels in different niches. *E. coli* and *Salmonella Typhimurium* have nearly identical GyrA and GyrB proteins and identical promoter sequences for essential genes. Nonetheless, *Salmonella* maintains an average chromosome supercoil density 15% lower than *E. coli*, and *E coli* cannot grow at the supercoil density maintained in *Salmonella* [46]. The solution to this paradox turned out to be determined by the C-terminal 35 and 38 amino acids of *Salmonella* and *E. coli* GyrA, respectively [47,48].

Martin was incredibly cautious and precise in his science, always testing or proving things via complementary approaches. But he also showed an enchanting and unique personality. When he visited Pat’s lab in 1989 to work on the SGS project, they took off on Sunday to stretch their legs on a 5-mile loop hike near the Appalachian Trail. It is an old Indian trail that starts near Birmingham and goes all the way to upper Canada. Near the end of their mini trek, Pat’s 8-year-old daughter, Rachel, “sprained” her ankle and convinced Martin to hoist her up on his shoulder on the way home, which he did with ease. She still remembers that day. He loved traveling. He had the amazing ambition to see all of Vermeer’s paintings and followed them wherever they happened to be displayed. He visited Afghanistan in the 1970s, and it was wonderful to listen to his memories while we traveled around Pakistan, during a memorable Mu workshop in Islamabad in 1984. But he would also enjoy sitting in his little stereo room, listening to the amazing Corky Siegel play blues harp with an accompaniment by the Denver Symphony Orchestra. He loved to drive his ancient BMW sports car to Boulder and the mountains, or through the plains or to a ski resort or a Keystone meeting. The car had a hole in the floorboard, big enough to see the pavement below, and fur flying out the windows from his beloved dog, Shanty, standing on the back seat. We loved waking up on crisp Colorado mornings to the smell of his French-press espresso and a croissant from the local walk-to bakery, or to eat in the evening at a close-by Italian or far-away Mexican restaurant. Martin (Figure 2) is sorely missed by friends, all over the world.

## Figures and Tables

**Figure 1 viruses-15-00637-f001:**
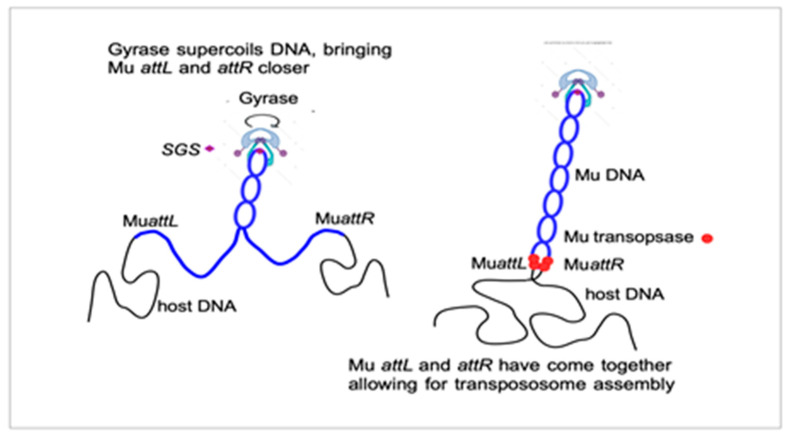
A very schematic view of how gyrase and the SGS could bring Mu ends together to interact with the pA transposase.

**Figure 2 viruses-15-00637-f002:**
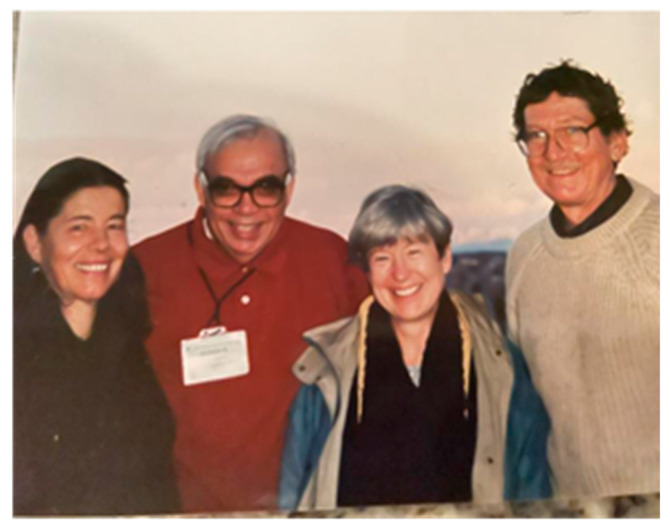
From right to left, Martin Pato, Geneviève Maenhaut, Nick Cozarelli and Ariane Toussaint (taken by Pat Higgins at a Keystone meeting in Santa Fe in the 1990s).

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
