# Peer review of "Twenty Years of Collaboration to Sort out Phage Mu Replication and Its Dependence on the Mu Central Gyrase Binding Site"

_viruses, 2023, doi:10.3390/v15030637_

Round 1
Reviewer 1 Report
This is a nice tribute to Martin Pato. I had several editorial comments, but due lack of line numbers to indicate these, I made edits directly in the PDF. I have tried to highlight the changed text, but could not indicate the deleted text. I have added sticky notes at many places. I began highlighting many hyphenated words that should not be, but realized that they were typos that would be eventually fixed, so eventually stopped doing that.
Other comments
1. Line 5 on page 3: 'later shown to be 'arm'' does not convey meaning. Perhaps 'later refered to' might make more sense?
2. Page 3, para beginning with In 1979, Jim Shapiro…could benefit from shortening. I suggest deleting the 7 lines I have highlighted, because they are difficult to understand and not essential. In the same para, why is mini in parenthesis toward the end?

Author Response
Response to Hiroshi Nakai's comments
HN: That is why I think Ariane and Pat should briefly mention Kiyoshi Mizuuchi’s soluble transposition system, which allowed strand transfer to be catalyzed in a defined system (perhaps citing Craigie et al., 1985, PNAS 82: 7570-7574) A defined system for the DNA strand-transfer reaction at the initiation of bacteriophage Mu transposition: protein and DNA substrate requirements.
References are added and text adapted, lines 144-47.
HN: and a brief summary of their finding about the role of DNA topology in sensing the relative orientation of the two Mu ends (Craigie and Mizuuchi, 1986, Cell 45:793-800). Role of DNA topology in Mu transposition: Mechanism of sensing the relative orientation of two DNA segments CellVol. 45Issue 6p793–800). Ref and text added lines 190-193, ref 48 and 49.
And following this, HN goes on :
Once this important principle is established and understood, then the innovative idea behind the SGS, separate from the mechanism of active transpososome assembly, would be more readily understood by nonspecialists. There are a few important aspects of this thinking that should be more explicitly established and stated. First, that mini-Mu’s lack the SGS but are able to transpose. Second is the thinking that the larger the Mu element, the probability of Mu ends being brought together in the proper configuration by DNA supercoiling becomes statistically less frequent in the absence of the centrally located SGS. (The second complete paragraph on p. 3 explains the reduced Mu replication that results with increased Mu size; however, a reader would wonder after reading the article whether this is due to the inserts in Mu causing the SGS to no longer be located in a central location. The thinking that the centrally located SGS and gyrase may play more and more of a critical role as the Mu ends are located farther and farther apart is never explicitly stated.) This helps to convey to nonspecialists that when the Mu ends in the proper orientation become far enough apart, the help of SGS and DNA gyrase can serve to organize the topological structure of Mu DNA to help bring together the ends in their proper configuration. The impact of this thinking and results was that it introduced a possible new function of topoisomerases in general and certain types of topisomerase binding sites. That thinking may be so apparent to the authors that they may not be considering the possibility that some readers may not fully comprehend that the most important mechanism being pointed out in Figure 1 is the role of the gyrase and the SGS in helping to bring distant sites together in the proper configuration (i.e., making it statistically more likely), not the fact that the ends brought together are providing the scaffold for active transpososome assembly (what was established in other studies). The fact that gyrase and SGS can play this organizing role is what I think Martin would have wanted the readers to understand completely and to attribute to his work (and thus the importance that this is especially clear). And this mechanism is what possibly has important ramifications for understanding chromosome function, the interaction between a variety of elements at distance, and its regulation, a topic that would be of interest to a wide variety of readers.
I is difficult to say we strictly comply to this long comment. As stated below, related to several of the Harshey's comments, several sections were rewritten trying to keep a simplification of the scientific complexity of some of the experiments, to better explain the general significance of Martin's scientific contribution and to keep the text easy to follow by a general audience!
Minor suggested edits: the inverse image of Greek lambda were indeed noticed:but we don't know how to correct it!
1. Last sentence of the second paragraph of page 2; better (edited portion in brackets): Transposable antibiotic resistance genes had been shown to move [from one replicon to another]... Done
2. First sentence of last paragraph on page 2: Mu ends and pA and pB[, the] transposase and transposition activator[, respectively,] are essential... Done
3.. Figure 1 legend: A very schematic view of how gyrase and the SGS could bring Mu ends together [in the proper orientation to enable the assembly of the active pA transpososme.] Done
4. Line 7 of very first paragraph (p. 1): period after world. Done
Response to R. Harshey's comments.
"In addition, many structures were seen, which contained two forks usually separated by a segment shorter than the length of the replicating mini-Mu, hence possibly partially replicated mini-Mu. To preserve such intermediates, which would have been destroyed by heteroduplex mapping, the mini-Mu was localized by its partial denaturation map, analyzing the correlation be-tween a standard denaturation map of the mini-Mu and the denaturation profiles of the various DNA structures observed after mini-Mu induction. "
This section has been replaced by lines 106-116 to improve clarity.
RH: This sentence might benefit from adding ', perhaps the pairing of the two Mu ends':
Done line 137
Lines 147-196 ("At a 1989 Gordon conference Pat learned an interesting.... to It suggests that the extrusion of a Mu-containing domain within the context of the bacterial nucleoid, helps bringing Mu ends together prior to transpososome assembly.") are the result of a rewriting to take into account several comments of RH on the section while keeping some simplification of the scientific complexity of these experiments. We hope it will now be easier to follow by a general audience.
Lines 201-216: here again we rewrote a section hoping to answer several of RH comments on our lack of clarity. In view of how she sent her comments in the pdf file with no line numbers we didn't see a way to respond comment by comment and hope it is OK this way.
Reviewer 2 Report
First of all, let me say that I think Martin Pato would have loved this, especially written by his long-time friends and colleagues. I remember his speaking to me a lot about the role of DNA gyrase and the Mu strong gyrase site (SGS) and their role in Mu DNA replication. The review tells an interesting story of how the collaboration between him, Pat, and Ariane progressed, and that is the heart of this review. I would not want to see any revisions that would significantly change the narrative describing this relationship between friends and colleagues. My only major suggestion is in helping even the nonspecialist reader fully understand what was innovative about this thinking about the SGS and its role in transposition, distinguished from other important concepts that were established by other investigators but that would be an important part of the overall picture of what SGS and gyrase are doing. That picture is well illustrated in Figure 1, which crystallizes the thinking about SGS. What is so important in understanding Martin’s legacy and the collaboration described in this article is just what part of the model is the innovative thinking by Martin and his colleagues and how this had significant impact in terms of thinking about topoisomerases, topoisomerase binding sites, and their function in general. The figure in part depicts the two Mu ends coming together to promote the assembly of an active oligomeric transpososome from transposase monomers. This part of the model represents a lot of work done by multiple investigators, summarized in the review cited toward the beginning of this article, but this specific aspect (i.e., the specific orientation of Mu ends promoting active transpososome assembly) should be distinguished from the other part of this overall picture that is derived from Martin’s work, which builds upon these works and correlates well with concepts established by them. That is why I think Ariane and Pat should briefly mention Kiyoshi Mizuuchi’s soluble transposition system, which allowed strand transfer to be catalyzed in a defined system (perhaps citing Craigie et al., 1980, PNAS 82: 7570-7574) and a brief summary of their finding about the role of DNA topology in sensing the relative orientation of the two Mu ends (Craigie and Mizuuchi, 1986, Cell 45:793-800). Once this important principle is established and understood, then the innovative idea behind the SGS, separate from the mechanism of active transpososome assembly, would be more readily understood by nonspecialists. There are a few important aspects of this thinking that should be more explicitly established and stated. First, that mini-Mu’s lack the SGS but are able to transpose. Second is the thinking that the larger the Mu element, the probability of Mu ends being brought together in the proper configuration by DNA supercoiling becomes statistically less frequent in the absence of the centrally located SGS. (The second complete paragraph on p. 3 explains the reduced Mu replication that results with increased Mu size; however, a reader would wonder after reading the article whether this is due to the inserts in Mu causing the SGS to no longer be located in a central location. The thinking that the centrally located SGS and gyrase may play more and more of a critical role as the Mu ends are located farther and farther apart is never explicitly stated.) This helps to convey to nonspecialists that when the Mu ends in the proper orientation become far enough apart, the help of SGS and DNA gyrase can serve to organize the topological structure of Mu DNA to help bring together the ends in their proper configuration. The impact of this thinking and results was that it introduced a possible new function of topoisomerases in general and certain types of topisomerase binding sites. That thinking may be so apparent to the authors that they may not be considering the possibility that some readers may not fully comprehend that the most important mechanism being pointed out in Figure 1 is the role of the gyrase and the SGS in helping to bring distant sites together in the proper configuration (i.e., making it statistically more likely), not the fact that the ends brought together are providing the scaffold for active transpososome assembly (what was established in other studies). The fact that gyrase and SGS can play this organizing role is what I think Martin would have wanted the readers to understand completely and to attribute to his work (and thus the importance that this is especially clear). And this mechanism is what possibly has important ramifications for understanding chromosome function, the interaction between a variety of elements at distance, and its regulation, a topic that would be of interest to a wide variety of readers.
Minor suggested edits (aside from the inverse image of Greek lambda that I’m sure the authors noticed:
1. Last sentence of the second paragraph of page 2; better (edited portion in brackets): Transposable antibiotic resistance genes had been shown to move [from one replicon to another]…..
2. First sentence of last paragraph on page 2: Mu ends and pA and pB[, the] transposase and transposition activator[, respectively,] are essential…..
3.. Figure 1 legend: A very schematic view of how gyrase and the SGS could bring Mu ends together [in the proper orientation to enable the assembly of the active pA transpososme.]
4. Line 7 of very first paragraph (p. 1): period after world.
Author Response
Response to R. Harshey's comments.
"In addition, many structures were seen, which contained two forks usually separated by a segment shorter than the length of the replicating mini-Mu, hence possibly partially replicated mini-Mu. To preserve such intermediates, which would have been destroyed by heteroduplex mapping, the mini-Mu was localized by its partial denaturation map, analyzing the correlation be-tween a standard denaturation map of the mini-Mu and the denaturation profiles of the various DNA structures observed after mini-Mu induction. "
This section has been replaced by lines 106-116 to improve clarity.
RH: This sentence might benefit from adding ', perhaps the pairing of the two Mu ends':
Done line 137
Lines 147-196 ("At a 1989 Gordon conference Pat learned an interesting.... to It suggests that the extrusion of a Mu-containing domain within the context of the bacterial nucleoid, helps bringing Mu ends together prior to transpososome assembly.") are the result of a rewriting to take into account several comments of RH on the section while keeping some simplification of the scientific complexity of these experiments. We hope it will now be easier to follow by a general audience.
Lines 201-216: here again we rewrote a section hoping to answer several of RH comments on our lack of clarity. In view of how she sent her comments in the pdf file with no line numbers we didn't see a way to respond comment by comment and hope it is OK this way.